# Machine-Learning-Based Detecting of Eyelid Closure and Smiling Using Surface Electromyography of Auricular Muscles in Patients with Postparalytic Facial Synkinesis: A Feasibility Study

**DOI:** 10.3390/diagnostics13030554

**Published:** 2023-02-02

**Authors:** Jakob Hochreiter, Eric Hoche, Luisa Janik, Gerd Fabian Volk, Lutz Leistritz, Christoph Anders, Orlando Guntinas-Lichius

**Affiliations:** 1Department of Medical Engineering, University of Applied Sciences Upper Austria, 4020 Linz, Austria; 2MED-EL Elektromedizinische Geräte GmbH, 6020 Innsbruck, Austria; 3Department of Otorhinolaryngology, Jena University Hospital, 07743 Jena, Germany; 4Facial-Nerve-Center, Jena University Hospital, 07747 Jena, Germany; 5Center for Rare Diseases, Jena University Hospital, 07747 Jena, Germany; 6Institute of Medical Statistics, Computer and Data Sciences, Jena University Hospital, 07743 Jena, Germany; 7Division for Motor Research, Pathophysiology and Biomechanics, Department for Trauma-, Hand- and Reconstructive Surgery, Jena University Hospital, 07743 Jena, Germany

**Keywords:** auricular muscles, facial muscles, human, facial palsy, electrophysiology, ear wiggling, muscle trigger, support vector machine

## Abstract

Surface electromyography (EMG) allows reliable detection of muscle activity in all nine intrinsic and extrinsic ear muscles during facial muscle movements. The ear muscles are affected by synkinetic EMG activity in patients with postparalytic facial synkinesis (PFS). The aim of the present work was to establish a machine-learning-based algorithm to detect eyelid closure and smiling in patients with PFS by recording sEMG using surface electromyography of the auricular muscles. Sixteen patients (10 female, 6 male) with PFS were included. EMG acquisition of the anterior auricular muscle, superior auricular muscle, posterior auricular muscle, tragicus muscle, orbicularis oculi muscle, and orbicularis oris muscle was performed on both sides of the face during standardized eye closure and smiling tasks. Machine-learning EMG classification with a support vector machine allowed for the reliable detection of eye closure or smiling from the ear muscle recordings with clear distinction to other mimic expressions. These results show that the EMG of the auricular muscles in patients with PFS may contain enough information to detect facial expressions to trigger a future implant in a closed-loop system for electrostimulation to improve insufficient eye closure and smiling in patients with PFS.

## 1. Introduction

The auricle of humans contains three extrinsic and six intrinsic muscles [1]. All ear muscles are innervated by the facial nerve [1]. Berzin and Fortinguerra showed EMG activity in the three external muscles (anterior, superior, and posterior auricular) during smiling and yawning [2]. Recently, Rüschenschmidt et al. established a standardized protocol for a reliable surface EMG examination of all nine ear muscles for a set of standard mimic movements, eye movement, and ear wiggling. In healthy participants, most tasks led to the activation of several ear muscles without any side difference [3]. The largest EMG activity in the auricular muscles was seen when smiling.

EMG activity disappears in the posterior auricular muscle, similar to mimic muscles in patients with acute facial palsy [4]. Furthermore, patients with postparalytic facial synkinesis as a sequela of pathological regeneration also show synchronous EMG activity in the posterior auricular muscle with the orbicularis oculi muscle during eye blinking. This was confirmed for all other auricular muscles, i.e., all ear muscles of the paretic side were involved in synkinetic activation [3].

Measurement of ear muscle EMG activity offers an unexploited potential for several medical monitoring or neuroprosthetic applications: Auricular muscle EMG allows for the monitoring of emotional states, brainstem lesion diagnostics, or of stroke manifestations [5]. Patients with quadriplegia can learn to steer a wheelchair with ear muscle activation [6]. Using EMG signals from healthy muscles in the face to trigger the stimulation of paralyzed muscles was investigated by Frigerio et al. [7]. They used the EMG activity of the contralateral orbicularis oculi muscle to trigger the electrostimulation of the same muscle on the paralyzed side. Surface EMG recordings from the contralateral side for the detection of blinking or smiling were also evaluated by Rantanen et al. [8]. They reported favorable results for the detection of smiles and puckering lips; however, they did have difficulties with blink detection caused by interfering facial expressions. Instead of using signals from the contralateral side, we already conducted a study based on signals measured by needle EMG within the zygomatic and the orbicularis oculi muscles of the paretic side. The algorithm used for classification, based on the shape detection of single muscle unit action potentials followed by the classification with a long short-term memory network, did deliver promising results [9]. For later use in a clinical setting, the approach will only be feasible if at least some degree of innervation is present in the target muscles. This innervation can come from remaining axons or regrown axons. Aberrant reinnervation will provide distinguishable activity. In contrast, such an approach will not be feasible in patients with permanent complete paralysis of the facial muscles. 

Therefore, the present study tries to evaluate the feasibility of using the auricular muscles as a trigger source for the detection of facial movements, as these could potentially still be intact even in patients with otherwise peripheral facial paralysis. We hypothesize that the synkinetic co-contractions in the auricular muscles will provide enough information to enable an EMG-based classification of facial movements. As Surface EMG was used, the formerly described algorithm based on single muscle unit action potentials could not be used. The present work describes a new approach with window-based feature extraction and classification with a support vector machine (SVM), starting with an evaluation in patients with postparalytic facial synkinesis.

## 2. Material and Methods

### 2.1. Study Design and Study Population

A prospective, observational feasibility study was performed. The study was approved by the ethics committee of the Jena University Hospital (No. 2018-1103-BO). All participants provided written informed consent. Sixteen patients (10 female, 50 ± 15 years; 6 male, 57 ± 15 years) with postparalytic facial synkinesis after acute unilateral (7 right side, 9 left side) peripheral facial paralysis (range of time since onset: 12–322 months) were examined.

### 2.2. Multi-Channel Surface Electromyography Setting and Facial Movement Tasks

A standard EMG system (Tower of Measurement, DeMeTec, Langgöns, Germany) and recorder (Biovision, Wehrheim, Germany; sampling rate: 4096 samples/second; amplitude resolution: 0.596 nV; low pass filter at 2 kHz.) was used. Small Ag/AgCl cup-electrodes (5 mm; GVBgeliMED, GmbH, Bad Segeberg, Germany) applied with electrode gel (CareFusion, Helsinki, Finland) and adhesive tape were used for the EMG acquisition of the following auricular muscles on both ears: anterior auricular muscle (AAM), superior auricular muscle (SAM), posterior auricular muscle, and tragicus muscle (TM) (Figure 1). Two facial muscles of the unaffected side of the face (orbicularis oculi muscle (OOcM) and orbicularis oris muscle (OOrM)) were recorded using larger Ag/AgCl self-adhesive, disposable electrodes (1.6 cm, H124SG, Covidien, Dublin, Ireland). These signals will later serve as the ground truth for data labeling. The two electrodes for the OOcM were placed at the lateral corner of the eye and medial below the OOcM. For the recording of the OOrM, the electrodes were at the corner of the mouth and 2 cm lateral of the philtrum. The reference electrode was placed on the mastoid process and the common ground on spinal process C7. The recordings were performed synchronously on both sides of the face (postparalytic side, contralateral side).

Participants sat in a relaxed upright position. The patients performed mimic movements according to a standardized protocol [10]. Closing the eyes (CE), smiling with lips closed (SLC), and smiling showing the teeth (ST) were the target exercises, i.e., the movements that were supposed to be recognized later by the algorithm. SLC and ST were considered one group for the classification. The other exercises were: face at rest, nose wrinkling, frowning, lip pursing, and clenching the teeth. These exercises were selected to provide homogeneous involvement of the different other facial muscles as so-called ‘interference exercises’. Each exercise was repeated five times, each repetition lasting for 3 s, separated by a 3-s face at rest period. This paradigm was the same for all exercises except for face at rest and CE. During face at rest, the patients relaxed all facial muscles for 20 s and for CE, closed their eyes continuously for 20 s. All face at-rest periods in the data were considered as belonging to the interference exercise group. 

### 2.3. Data Processing and EMG Data Labeling

Data processing was performed with Python 3.6 (Python Software Foundation, Wilmington, DE, USA). The data were band pass filtered from 1 Hz to 1 kHz and notch filtered at 50 Hz (2nd Order, Butterworth). The unconventionally high upper cut-off frequency of 1 kHz was chosen because the auricular muscles have only a thin amount of skin coverage compared to other skeletal muscles. This reduces the low pass effect of the tissue between the muscle and the electrode; therefore, the frequency range of the EMG is increased. For all muscles equipped with two electrodes (cf. Figure 1), bipolar signals were calculated from the respective electrode pairs to reduce cross-talk, i.e., to obtain the EMG data of the individual muscles. There was only one electrode placed on the TM; this particular channel was further used in its original monopolar montage. The EMG data were labeled manually with the starting and end points of the different facial exercises based on the recorded facial muscles on the unaffected side of the face. Activation of the OOcM was defined as the gold standard for the target exercise CE. The activation of the OOrM was the standard for the target exercises SLC and ST. Activation of both muscles was the reference for the interference exercises. Muscle activation was defined as a clear amplitude increase in the EMG signal of the respective muscle. As the recording paradigm was designed such that each facial movement was separated by a 3-s rest period, a well-defined onset was visible in most recordings. Inconclusive cases or EMG activity which was not in line with the experimental paradigm were excluded.

### 2.4. EMG Classification with a Support Vector Machine (SVM)

For the machine learning EMG classification, a set of 10 different well-established EMG features per muscle was calculated and normalized from a moving window with 50% overlap. We compared three different window lengths (33, 66, and 100 ms). All features and the equation for normalization are summarized in Appendix A. An SVM configurated with a Gaussian Radial Basis Function Kernel was implemented using the scikit-learn software package [11] to classify the individual windows as belonging either to the class CE, SLC/ST, or Interference. Due to the random nature of the synkinetic reinnervation process, each patient presented a unique, fingerprint-like EMG response in the auricular muscles. To deal with this unique EMG activity across patients, the SVM was trained for each patient individually. As this study shall demonstrate the feasibility of using the described method as a trigger for a neuroprosthetic application, it is necessary to keep the system complexity as low as possible. Thus, the number of EMG channels available for classification was limited to one or two muscles. Which muscle to use had to be decided on a patient basis due to the uniqueness of the EMG activity. In order to reduce the dimensionality of the feature space when using two muscles, 10 out of the 20 available features (10 per muscle) were selected utilizing Mutual Information between the labels and features [12,13]. When only a single muscle was used, all 10 features were included in the classification. The data set was split into a 2⁄3 training data set and a 1⁄3 test data set. To avoid dependencies between training and test data possibly caused by overlapping windows, data was always split into time-wise continuous thirds. The absolute number of data available is listed in Table 1. To find the best-performing muscle or muscle pair and the adjacent SVM-Hyper-Parameters (C and γ), a grid search was performed for each patient based on the training data. For C and γ, a logarithmic search space (C: 10^−1^ to 10^3^ and γ: 10^−4^ to 10^2^) was used, and 5-fold cross-validation was performed within the test data to find the best parametrization. Once the best-performing muscle/muscle pair with their hyperparameters was found, three-fold cross-validation of the model was performed using this parametrization.

To provide an objective measure of the performance of the classifier, despite the imbalanced data set (Table 1), the macro F1-score was calculated. This score provided the relationship between the number of windows truly belonging to a certain class and the number of windows classified as belonging to it on a per-class average. It can reach values from 0 (worst) to 1 (best). In addition, the per-class F1-score was used to analyze the performance of the individual classes. This score is similar to the macro F1-Score; however, it is not based on a per-class average and thus provides a score for each class [14]. The scores finally reported for each patient represent the mean value of the three results from the three-fold cross-validation. To demonstrate the increased information content of the synkinetic auricular activity, the algorithm was also tested on the data recorded from the contralateral side (i.e., healthy side) and compared to the results achieved on the postparalytic side.

### 2.5. Statistical Analysis

The difference in the median macro F1-score between the three different window lengths was analyzed with the Kruskal–Wallis test. The difference in the median macro F1-score of the groups with one or two muscles, as well as the difference between the paretic and the c side contralateral side, was analyzed with a Mann–Whitney U-Test. The difference in the median per-class F1-score was also analyzed with the Kruskal–Wallis test followed by the Conover post hoc test with a *p* adjustment according to the Holm–Bonferroni method [15]. For all statistical tests, the significance was two-sided and set to *p* < 0.05.

## 3. Results

### 3.1. Optimal Window Lengths for Algorithmic Classification

The window lengths 33 ms, 66 ms, and 100 ms were compared to analyze the optimal parametrization of the proposed algorithm. The comparison is depicted in Figure 2. The performance of the algorithm, i.e., the macro F1-scores, was not significantly different between the three windows (all *p* > 0.05). Thus, further analysis was performed using a 66 ms window, seen as a tradeoff between time responsiveness and computational expense.

### 3.2. Determination of the Optimal Auricular Muscle EMG Recording Setting

To determine whether the algorithm performs better with the EMG of one or two auricular muscles as an input, the median of the macro F1-score was compared for both cases (Figure 3). The macro F1 score was significantly higher (*p* = 0.017), i.e., the algorithm performed better, when the EMG recording data of two auricular muscles were included instead of using the information available from one auricular muscle. The best-performing muscle/muscle pair varied between patients without any systematic preference. Table 2 details the chosen muscle pair as well as the hyperparameters for each patient in the best-performing algorithm parametrization (wl = 66 ms, two muscles). It also lists the selected features for each patient order in accordance with mutual information. It shall be noted that features 8, 9, and 10 are never chosen when two muscles are available for classification. However, they increase the algorithm performance when only one muscle is available for classification.

### 3.3. Comparison of the per Class Macro F1-Scores between the Face at Rest and the Activation Tasks

For a better understanding of the performance of the algorithm, the per-class macro F1-score for a window length of 66 ms and two muscles was analyzed (Figure 4). The classification performance of the algorithm was different between the interference exercises and the target exercises CE or SLC/ST (*p* < 0.001). The posthoc test revealed a significant difference in the interference exercises and CT (*p* < 0.001) as well as between the interference exercises and SLC/ST. (*p* < 0.001). There was no statistical difference in the macro F1-score between the tasks CT versus SLC/ST (*p* = 0.776).

### 3.4. Comparison of the Auricular Muscle Activation between the Paretic and the Contralateral Side

Figure 5 illustrates the comparison between the performance of the algorithm on the auricular EMG of the contralateral and the postparalytic side. The macro F1-scores were significantly higher on the postparalytic side (*p* = 0.007).

## 4. Discussion

The clinical potential to use ear muscle EMG for facial diagnostics has been neglected so far. Recently, we could show that surface EMG is reliably feasible from nearly all extrinsic and intrinsic ear muscles [3]. The ear muscle EMG recordings showed distanced patterns during specific facial movements, especially very strong during smiling. Furthermore, the ear muscles also show strong synkinetic activity in patients with PFS. In the present study, we show that these distinct EMG patterns are characteristic to such an extent that they can be used to classify facial movements with a machine-learning-based algorithm. When fitted individually to each patient, the algorithm could detect eye closure and smiling from extrinsic ear muscle EMG recordings. This offers new possibilities for a closed-loop stimulation system for facial muscle pacing.

Tobey and Sutton, as well as Rothstein and Berlinger, were the first to show in rabbits that EMG signals from facial muscles can be used to trigger electrostimulation of denervated muscles on the other side of the face [16,17]. Kurita et al. were the first to demonstrate in an experimental setting that it is theoretically possible to trigger an electrostimulation of a paralyzed frontalis muscle by simultaneous EMG recordings of the contralateral frontalis muscle while frowning [18]. Hereafter, Griffin and Kim published the idea to use an implantable electric prosthesis system to realize such a closed-loop system [19]. Later on, it was shown by Frigerio et al. that both surface EMG of the orbicularis oculi muscle and also infrared blink detection glasses allow reliable blinking detection for a closed-loop facial pacing system [20,21]. Jowett et al. showed in rats that the stimulation could also be performed with cuff electrodes around facial nerve branches [22]. Recently, Hasmat et al. proposed to use of the contralateral trigger to steer an electromagnetic actuator in combination with an eyelid sling to restore eyelid function [23]. We are convinced that the ear muscles of the paralyzed side have very attractive potential as a trigger source. No surgery on the contralateral healthy side is needed to avoid side effects or complications on the healthy side. In patients with peripheral facial nerve damage beyond the main trunk, the auricular facial branches to the ear muscles might be unaffected, providing an optimal ipsilateral trigger source for patients with denervated facial muscles. Furthermore, in patients with PFS, such as in the present study, the ear muscles are not the target muscles for rehabilitation therapy, making the ear muscles, otherwise functionally unimportant in humans, a useful functional diagnostic and therapeutical tool.

The median macro F1-score, used as a parameter of the classification performance in the present study, was 0.850. Leistritz et al. used needle EMG, i.e., intramuscular EMG, as a signal and revealed a macro F1-score of 0.90 in six patients using a long short-term memory (LSTM) network for classification [9]. Frigerio et al. also used surface EMG but only from the orbicularis oculi muscle and blink triggering [18]. They revealed a macro F1-score of 0.89. This comparison shows that EMG recordings from the ear muscles perform approximately equally to the range obtained by the recording of facial muscles.

Upon inspection of the per class F1-score (Figure 4), one could perceive that the algorithm performs worse in the detection of CE compared to SLC/ST. However, no statistically significant difference could be proven between the two (*p* = 0.776). Nevertheless, the larger IQR of CE compared to SLC/ST could indicate that the CE exercise is more difficult to detect in the auricular muscles. One reason might be that smiling is among the facial movements that elicit the most auricular EMG activity in patients and healthy subjects [3]. Individual differences in the synkinetic activation pattern also play a role in this respect. Improvements in classification performance could be achieved by implementing an (LSTM) network as a classifier either in combination with a greater window overlap and, therefore, a higher decision rate or without any feature extraction, applied directly to the raw data [24]. Compared to an SVM, an LSTM network is capable of learning sequence dependencies which could aid in improving the performance [25,26]. However, this approach would either be computationally very expensive due to the calculation of more features per second or would require a larger data set for training to be available.

When comparing the performance of the algorithm on the auricular EMG of the healthy side to the postparalytic side, it performed significantly better on the latter. This is in line with our hypothesis that the synkinetic reinnervation of the auricular muscles adds additional information about facial movements to the auricular muscle activity.

A limitation of the current study was that only patients with PFS were included. Therefore, it remains to be shown that the proposed approach is also feasible in chronic facial palsy patients with permanent denervation of the mimic muscles but restored function of the ear muscles. This evaluation should be one of the goals of future investigations. Another aspect that would need to be investigated is the performance of the algorithm in a more realistic setting. In a real-world use scenario, the patient would often perform different facial expressions at the same time, e.g., closing the eyes while smiling. This could lead to a less well-defined EMG activity and would need special attention in the algorithm development. As a whole, the present study does provide valuable contributions in the steps towards the development of a closed-loop bionic implant for facial muscle stimulation in patients with different forms of facial palsy. Beyond applications in the rehabilitation of facial palsy, EMG recording of ear muscles should be considered an interesting spot for the human–machine interface in the future.

## 5. Conclusions

Surface EMG recordings of the ipsilateral extrinsic ear muscles allowed the development of a reliable support vector machine-based EMG classification of the ear muscle signals to detect eye closure and smiling in patients with PFS. The algorithm was fitted individually to each patient to best utilize the characteristic synkinetic activity in the auricular muscle. This signaling could be used as a trigger in a purely ipsilateral closed-loop system in a future bionic implantable medical device for electrostimulation to improve insufficient eye closure and smiling in patients with PFS.

## Figures and Tables

**Figure 1 diagnostics-13-00554-f001:**
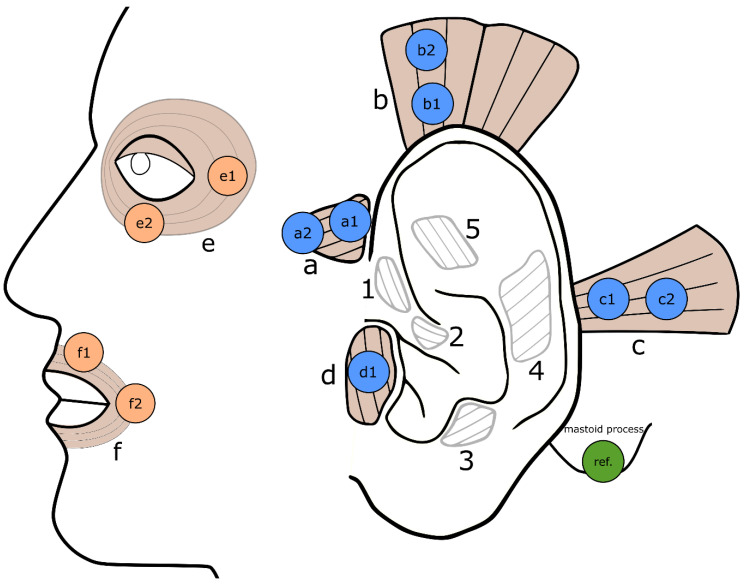
Electrode positioning on the auricular and mimic muscles. a = anterior auricular muscle (AAM); b = superior auricular muscle (SAM); c = posterior auricular muscle (PAM); d = tragicus muscle (TM); e = orbicularis oculi muscle (OOcM); f = orbicularis oris muscle (OOrM); 1–5: other auricular muscles, which are not relevant for this study. Electrodes are placed on auricular muscles (blue circles) and mimic muscles (orange circles). A reference electrode (green circle) is placed on the mastoid process; common ground is placed on the spinous process of C7 (not depicted). Electrodes were placed symmetrically on both sides.

**Figure 2 diagnostics-13-00554-f002:**
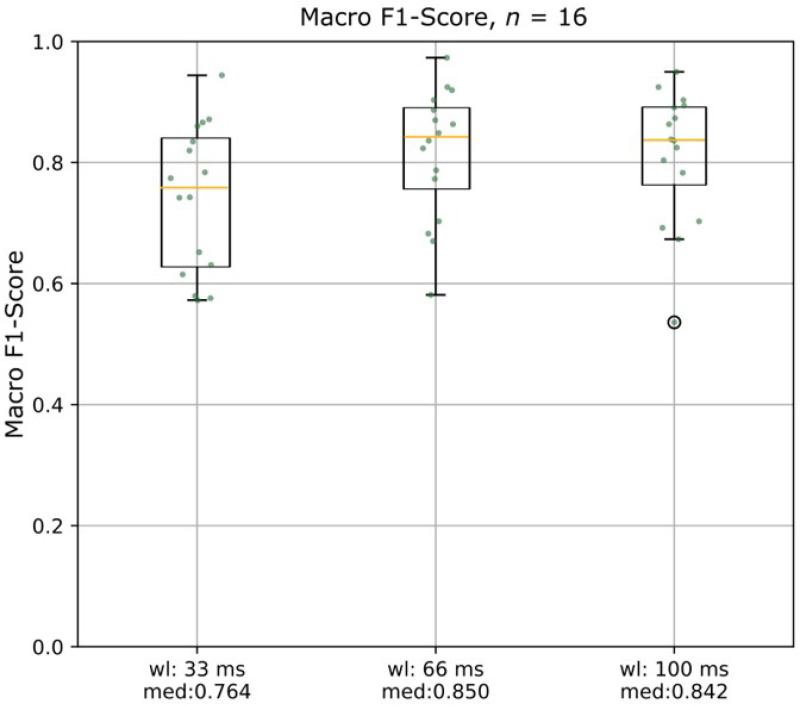
Boxplots of the comparison of the macro F1-scores of different window lengths (33 ms, 66 m, and 100 ms) using the two best-performing muscles in each patient. There was no significant difference between the three chosen window lengths. wl = window length; med = median macro F1-score.

**Figure 3 diagnostics-13-00554-f003:**
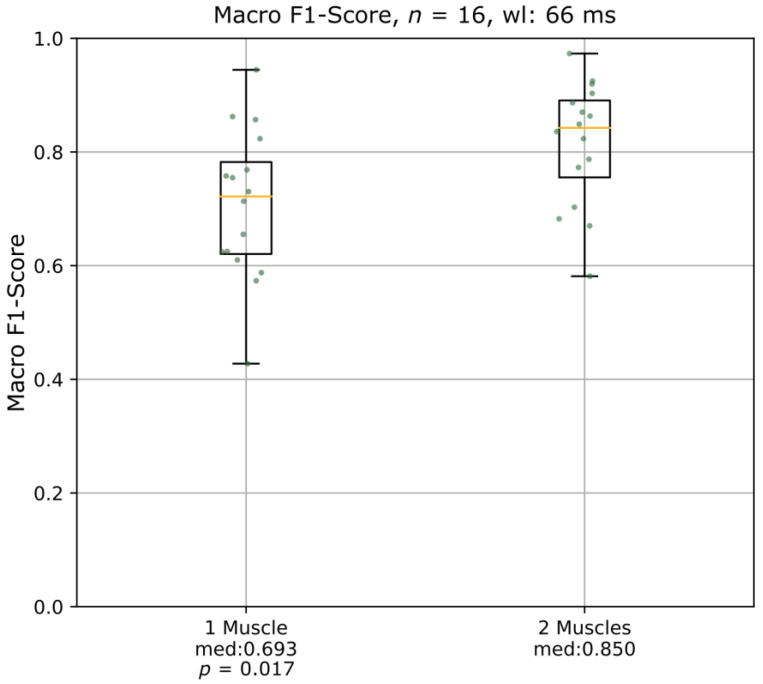
Boxplots of the comparison of the effect on the macro F1-scores using the electromyography of one muscle or of two muscles with a window length of 66 ms. wl = window length; med = median macro F1-score.

**Figure 4 diagnostics-13-00554-f004:**
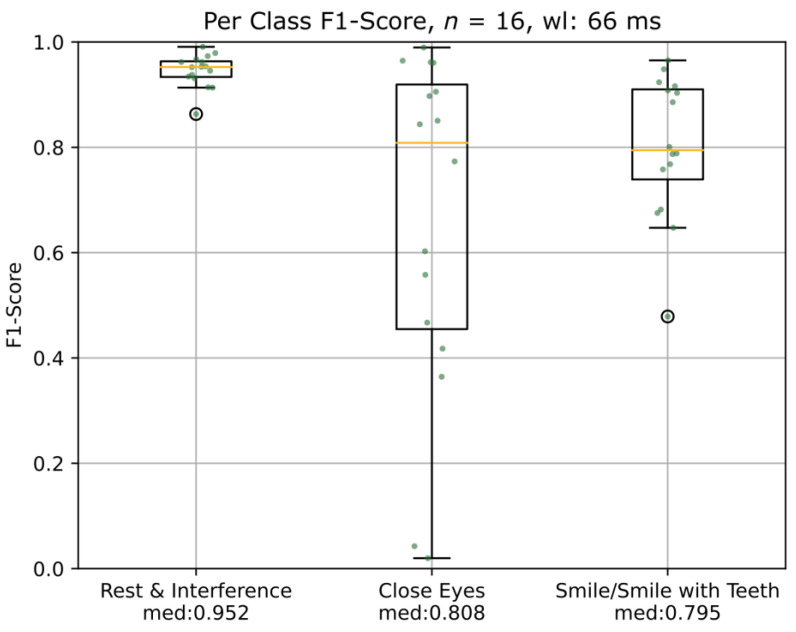
Boxplots of the comparison of the results of the F1-scores for the different tasks using the two best-performing muscles in each patient. wl = window length; med = median macro F1-score.

**Figure 5 diagnostics-13-00554-f005:**
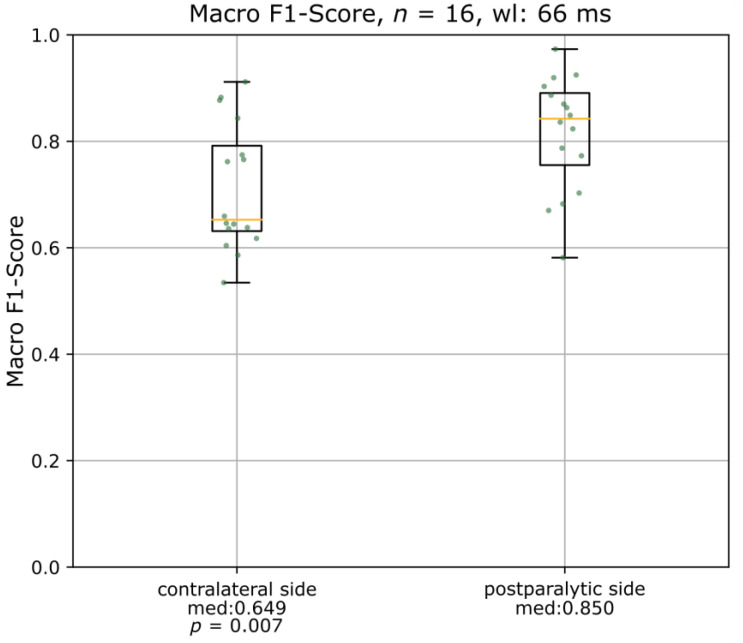
Boxplots of the comparison of macro F1-scores on the contralateral and the paretic side using the two best-performing muscles in each patient. wl = window length; med = median macro F1-score.

**Table 1 diagnostics-13-00554-t001:** Information regarding absolute amount of data available per class.

Number of Windows, WL = 66 ms	ECE	S and ST	Clenching the Teeth	Frowning	Lip Pursing	Nose Wrinkling	Face at Rest
Mean	293.4	2109.8	685.1	691.5	713.4	687.4	4356.4
Standard diviation	24.8	145.9	89.3	58.7	39.3	44.0	541.1
Minimum	215	1899	615	596	649	598	3231
Maximum	332	2503	981	804	791	779	5142

Values are calculated from windows available at window length = 66 ms and 50% overlap. WL = window length.

**Table 2 diagnostics-13-00554-t002:** Model parametrization with the best performance (wl = 66 ms, two muscles). Selected features are reported with reference to Appendix A. Order given is according to mutual information ranking.

Patient	Selected Muscles	γ	C	Macro F1-Score	Selected Features
1	PAM, TM	0.010	1000.0	0.841	TM1, TM4, PAM1, TM2, PAM2, PAM5, PAM7, TM7, PAM4, TM5
2	SAM, PAM	0.001	1000.0	0.879	PAM5, PAM2, SAM5, PAM4, SAM4, PAM1, SAM7, PAM7, SAM1, SAM2
3	SAM, AAM	0.010	1000.0	0.641	AAM1, AAM4, SAM4, SAM1, AAM2, SAM2, SAM7, AAM7, SAM5, AAM5
4	SAM, PAM	0.100	100.0	0.931	SAM4, SAM2, PAM2, SAM5, PAM4, PAM1, SAM7, PAM7, SAM1, PAM5
5	SAM, PAM	0.100	1.0	0.985	SAM5, SAM1, SAM7, SAM2, PAM2, PAM4, SAM4, PAM7, PAM1, PAM5
6	SAM, TM	0.100	10.0	0.736	SAM4, SAM2, TM2, SAM5, TM4, TM1, SAM7, TM7, SAM1, TM5
7	TM, AAM	1.000	100.0	0.723	TM1, TM4, TM2, AAM2, TM5, AAM4, TM7, AAM7, AAM1, AAM5
8	SAM, AAM	0.001	1000.0	0.817	AAM1, AAM2, AAM4, SAM2, SAM5, SAM4, SAM7, AAM7, SAM1, AAM5
9	SAM, PAM	0.001	1000.0	0.863	SAM1, SAM2, PAM2, SAM5, PAM4, SAM4, SAM7, PAM7, PAM1, PAM5
10	SAM, AAM	0.100	10.0	0.658	AAM6, AAM7, SAM4, AAM1, AAM4, SAM2, SAM7, SAM5, SAM1, AAM2
11	SAM, AAM	0.100	10.0	0.496	AAM1, SAM1, AAM2, AAM4, SAM2, SAM5, SAM7, AAM7, SAM4, AAM5
12	SAM, TM	10.00	1000.0	0.860	TM1, TM4, TM2, SAM5, SAM7, TM5, SAM4, TM7, SAM1, SAM2
13	PAM, TM	0.010	1000.0	0.770	PAM4, PAM2, TM2, PAM5, TM4, TM1, PAM7, TM7, PAM1, TM5
14	PAM, AAM	0.001	1000.0	0.884	AAM4, AAM2, AAM7, PAM1, AAM1, AAM5, PAM7, PAM5, PAM4, PAM2
15	SAM, PAM	0.100	10.0	0.893	SAM7, SAM2, PAM2, SAM5, PAM4, PAM6, PAM1, PAM7, PAM3, PAM5
16	SAM, PAM	0.001	1000.0	0.934	SAM1, SAM2, PAM2, SAM5, PAM4, SAM4, SAM7, PAM7, PAM1, PAM5

AAM = anterior auricular muscle; PAM = posterior auricular muscle; SAM = superior auricular muscle; TM = tragicus muscle.

## Data Availability

The datasets used during the current study are available from the corresponding author upon reasonable request.

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
