# Peer review of "Machine-Learning-Based Detecting of Eyelid Closure and Smiling Using Surface Electromyography of Auricular Muscles in Patients with Postparalytic Facial Synkinesis: A Feasibility Study"

_diagnostics, 2023, doi:10.3390/diagnostics13030554_

Round 1

Reviewer 1 Report

The paper reports a classification scheme based on an SVM using sEMG signals from patients with post-paralytic facial synkinesis. The problem statement is well-defined and they report the construction of a dataset with 16 patients. The results are focussed on some comparisons to determine the best window length, the use of one or two muscles, between others. 

There are some major concerns to be improved especially in the methodological approach.

1. When you used a machine learning method in biomedical data, it is necessary to report not only the model used, you need to report the hyperparameter set if you apply some kind of step for hyperparameter tuning, etc. Do you apply some kind of cross-validation?  

In the paper, the authors only said that they used an SVM, but they did not say anything about the model setting or the way to validate the model.

I recommend following the recommendation by Foster, Koprowski, and Skufca at https://www.ncbi.nlm.nih.gov/pmc/articles/PMC4105825/.

2. It is important to report the number of windows used in the training and test dataset per class. This information gives information about the balance of the classes, which is important to interpret the results.

3. There are some inaccuracies in the definition of the features in table 1. The equations for ZC and WL are not written according to the references. Phinyomark used a threshold in the computation of ZC and WL. Review the equation definition or specify if you did not use the Phinyomark definition to compute these features.

4. It is important to define the criteria used in the manual label process.

5. Why do you select a bandpass filter between 0.001Hz to 1Khz? Usually, EMG signals are bandpass between 10Hz and 500Hz.

6. It is better if you also report the F1 score per class.

7. It is important to be clear in the discussion and conclusion that your model is patient-specific (intrapatient). I understand that you need to train the model in each patient, which is a limitation for future applications.

8. There are other limitations in this study, there is not any feature selection scheme, there is not a hyperparameter tunning step, and there is no cross validation process.

Reviewer 2 Report

The manuscript entitled "Machine-learning based detecting of eyelid closure and smiling using surface electromyography of auricular muscles in patients with postparalytic facial synkinesis: a feasibility study" aims to establish a machine-learning based algorithm to detect eyelid closure and smiling in patients with PFS by a recording of sEMG using surface electromyography of auricular muscles. The title is long but very clear. To do it, the authors have data from 16 patients with postparalytic facial synkinesis after acute unilateral, peripheral facial paralysis. 

The manuscript is well-structured, but there are some problems that should be fixed.

1. Authors should explain how much data they have. For example, they explain they split the dataset in 2/3 to train and 1/3 to test, but we don't know the volume of data used to train these models.

2. Related to the fine tuning of the models, they must use methods like GridSearchCV or RandomSearchCV. Authors should add this kind of tests to their experiments.

3. To guarantee the reproducibility of the manuscript and the validity of the results, it is also essential that authors share the code realized to replicate the experiments and share it with the scientific community.

4. Authors should test SVM using cross-validation.

5. It is important to discuss these results with other research related to this subject using AI models.

Minor:

1. In the abstract, we can read: "Sixteen patients (10 female; median 56 years)" but I think that if they write the information about female patients, they should write also the information about men.

Round 2

Reviewer 1 Report

The manuscript has improved according to the corrections, however It still has the following issues:

In section 2.4 authors wrote:

a. "In order to reduce dimensionality of the feature space when using two muscles, 10 out of the 20 available features (10 per muscle) were selected utilizing Mutual Information between the labels and features." - Results of mutual information (which features were selected) must be reported in the results.

b. "To find the best performing muscle or muscle pair and the adjacent SVM-Hyper-Parameters (C and γ), a grid search was performed for each patient based on the test data." - Usually the grid search is performed divided the train set in a train and validation sets, or performing a cross validation in the training set. Test set must be used for a final test with the final model. Additionally, selected parameter must be reported in the results.

Section 3.2 is not clear enough. Signals were recorded in 5 muscle groups. However, in section 3.2 you only evaluate one or two auricular muscles as an input. What happen with the other information? Why did you select only two muscles? What muscles did you selected?

In section 3.2 authors wrote "The best performing muscle/muscle pair varied between patients". It is necessary to write more information about which muscle/muscle pairs were selected or which muscle/muscle pairs appear with more frequency. It is impossible to reproduce the results if the muscle/muscle pair is different in each patient and we do not have information. It is better if you select the pair with better performance and analyze the results in that pair for all the subjects.

 Figure 4 shows that for the close eyes task you have several results with the F1 scores less than 50%. It is necessary to discuss about these results. In discussion section the authors claim that "When fitted individually to each patient the algorithm could detect eye closure and smiling from extrinsic ear muscle EMG recordings.", however, results shows that the algorithm perform better to differentiate rest vs interference, and smile/smile teach. However, the results are very scatter for close eyes.

 It is not clear the purpose of section 3.4 Comparison of the auricular muscle activation between the paretic and the contralateral side. I understand that the muscles acquired in the contralateral and the postparalytic side are different (and the electrodes are different).

 In the discussion the authors claim: "This comparison shows that EMG recordings from the ear muscles perform approximately equally to the range obtained by the recording of facial muscles". However, it is not clear which pair of muscles were used in the results.

Author Response

Point-by-point response

Hochreiter et al. Machine-learning based detecting of eyelid closure and smiling using surface electromyography of auricular muscles in patients with postparalytic facial synkinesis: a feasibility study

Thank you very much for the second round of detailed reviews. We would like to answer to the queries of the reviewers point-by-point.

Reviewer #1

In section 2.4 authors wrote:

  1. "In order to reduce dimensionality of the feature space when using two muscles, 10 out of the 20 available features (10 per muscle) were selected utilizing Mutual Information between the labels and features." - Results of mutual information (which features were selected) must be reported in the results.

Answer 1.1:

Added Table 2

  1. "To find the best performing muscle or muscle pair and the adjacent SVM-Hyper-Parameters (C and γ), a grid search was performed for each patient based on the test data." - Usually the grid search is performed divided the train set in a train and validation sets, or performing a cross validation in the training set. Test set must be used for a final test with the final model. Additionally, selected parameter must be reported in the results.

Answer 1.2:

Thank you for pointing this out. This in unfortunately a typo. Of course, the trainings data was used for the Gridsearch. In detail: The sklearn.model_selection.GridSearchCV Method was used to perform 5-fold CV within the trainings data. à Corrected in Section 2.4. Hyper parameters are now state in Table Table 2

Section 3.2 is not clear enough. Signals were recorded in 5 muscle groups. However, in section 3.2 you only evaluate one or two auricular muscles as an input. What happen with the other information? Why did you select only two muscles? What muscles did you selected?

Answer 1.3:

*Signals where recorded in 4, not 5 auricular muscle groups (Figure 1)

Reduced number of Muscles is for reduction of system complexity for a potential neuroprosthetic pacing device. à Clarified in Section 2.4. (Also see answer 1.4 for additional information on which muscles where selected).

In section 3.2 authors wrote "The best performing muscle/muscle pair varied between patients". It is necessary to write more information about which muscle/muscle pairs were selected or which muscle/muscle pairs appear with more frequency. It is impossible to reproduce the results if the muscle/muscle pair is different in each patient and we do not have information. It is better if you select the pair with better performance and analyze the results in that pair for all the subjects.

Answer 1.4:

Synkinetic reinnervation is a process where nerve fibers of the damaged nerve partially do not grow back into their muscle of origin but also into other muscles (e.g. Former Zygomatic fibers partially grow into auricular muscles) à This process is non-deterministic and thus creates a unique EMG activity in each patient due to co-contractions (e.g auricular muscles also show activity when Zygomaticus is activated).

Therefore, it is not possible to use the same muscle pair for all patients and the algorithm has to be individually fitted to perform best.

In addition, in case of a pacemaker development, individualization of the implant is necessary anyway, thus it is not the goal of finding a “one fit’s all” approach, rather an adaptable algorithm. à Tried to clarify that in the end of Section 1,  in Section 2.4 and Section 3.2, also added the selected muscles in Supplemental Table 2.

Figure 4 shows that for the close eyes task you have several results with the F1 scores less than 50%. It is necessary to discuss about these results. In discussion section the authors claim that "When fitted individually to each patient the algorithm could detect eye closure and smiling from extrinsic ear muscle EMG recordings.", however, results shows that the algorithm perform better to differentiate rest vs interference, and smile/smile teach. However, the results are very scatter for close eyes.

Answer 1.5:

Added in Discussion.

It is not clear the purpose of section 3.4 Comparison of the auricular muscle activation between the paretic and the contralateral side. I understand that the muscles acquired in the contralateral and the postparalytic side are different (and the electrodes are different).

Answer 1.6:

Same electrodes where used, clarified in Section 2.2.

Purpose added at the ends of Sections 1, 2.3 and 4.

In the Discussion the authors claim: "This comparison shows that EMG recordings from the ear muscles perform approximately equally to the range obtained by the recording of facial muscles". However, it is not clear which pair of muscles were used in the results.

Answer 1.7:

Explained in answers above and added Table 2

To the Editorial Team

We planned the table with the formulas as Supplement Table 1. You formatted the manuscript in the MDPI Diagnostics style. Here, this table was integrated like a normal table, not as supplement. We can do this, but then the tables have to be re-numbered. Hence, supplement table 1 would be table 1, table 1 would be table 2, and the new table 2 would be table 3!

Jakob Hochreiter and Orlando Guntinas-Lichius

For all authors

Jena, 27-JAN-2023

Reviewer 2 Report

The authors have taken my suggestions on board. 

I believe that studies where there is code in experimentation should be shared and not doing so is against open science.

Round 3

Reviewer 1 Report

The authors responded adequately to the recommendations given in the previous reviews. Clarity regarding results has improved.